# Beyond Exit Code 1: Execution Report of RML Processors

Jakub Duchateau[1], Dylan Van Assche[2] and Christophe Debruyne[1]

[1]*Montefiore Institute, University of Liège, Belgium*
[2]*Independent Researcher, Brussels, Belgium*

## Abstract
The formalization of the RDF Mapping Language (RML) has led the KGC community to propose challenge tracks focused primarily on compliance coverage and performance, where compliance coverage was limited to "pass/fail," with test batteries. The formalization of RML benefited from community building and gathered insights from these tracks. While this drove the development of RML and its implementation, the focus on compliance and performance led to Developer Experience lagging behind. Current RML processors function as opaque black boxes, where the development lifecycle is reduced to a "Write-Run-Fail" loop, which is known to be frustrating and creates a barrier for uptake. Such barriers hinder the adoption of RML by non-experts. The issue is that there is no meaningful representation of the various errors for authoring and executing RML, which could aid agents (both human and computer-based, such as RML authors and RML IDEs), in engaging in meaningful ways with RML (processors). To bridge this gap, we propose the RML Execution Report (RER), a formalised, machine-readable artefact designed to facilitate such exchange by making issues transparent, structured, and meaningful. By interlacing execution statistics, plans, and a structured error taxonomy, we aim to transform the opaque `exit code 1` used in many processors into actionable feedback. We present RER to the community not as a finalised artefact, but as a foundation for discussion on improving the RML ecosystem.

## Keywords
Knowledge Graph Construction (KGC), RDF Mapping Language (RML), Error reporting

## 1. Introduction

The evolution of RML processors has been driven by the Knowledge Graph Construction Workshops (KGCW) and the challenges they pose. These benchmarks have mostly prioritised performance (in 2019 and 2023) and specification conformance (in 2021, 2024, and 2025).[1] While this focus has matured RML specifications and optimisation techniques for processors, the Developer Experience (DX) [1] around RML mapping feedback has not seen much development. This ties to the observation that a recurring complaint from KG construction practitioners is that tooling is inadequate.

In practice, the development lifecycle is reduced to a "Write-Run-Fail" loop in which RML processors provide binary success/failure signals, opaque stack traces, or processor-specific error messages. Without understandable, actionable feedback, the user is forced to spend a significant amount of cognitive effort debugging where the error lies and how to address it (which, as some of us have experienced, may involve informed guesses). We argue that the lack of observability (of the RML processing) and feedback raises the entry barrier for non-experts, as debugging may require knowledge of the processor's implementation rather than just the RML language or the metaphor used by the non-expert to author their mappings, thereby slowing broader adoption. RML was conceived as a declarative language, meaning that RML authors describe what should be transformed, and now how. This idea holds when everything goes well, but why should the processor not yield a meaningful result when the mapping goes wrong?

The lack of a "standardised"[2] reporting format leaves processors implementing disparate reporting strategies, that are often not machine-readable, hindering the creation of unified tooling. RML-Core [2]

*KGCW'26: 7th International Workshop on Knowledge Graph Construction, May 10, 2025, Dubrovnik, HRV*

✉ Jakub.Duchateau@uliege.be (J. Duchateau); work@dylanvanassche.be (D. Van Assche); C.Debruyne@uliege.be (C. Debruyne)

🆔 0009-0009-5090-8192 (J. Duchateau); 0000-0002-7195-9935 (D. Van Assche); 0000-0003-4734-3847 (C. Debruyne)

[1]There was no workshop in 2020 due to COVID19.
[2]When using "standardised", we are referring not to a standard, but to a community-agreed representation or way of doing things.

partially specifies the handling of "Data Errors" (Section 4.3), which requires processors to report invalid generated RDF terms. However, Section 4.3 addresses only a subset of potential issues (primarily focused on the generation of invalid RDF terms) and leaves the reporting mechanism implementation-defined.

We propose the RML Execution Report (RER) as a formalized, machine-readable artefact that not only considers RML-Core's Section 4.3 on Data Errors but also extends it to include execution, mapping, and configuration errors. By providing a vocabulary, RER enables RML processors to report errors with structured contextual metadata in a uniform and standardised way, yielding clearer, comparable, and actionable feedback rather than the opaque `exit code 1`. It also directly benefits RML tooling ecosystem including editors, validators, and CI pipelines that can now rely on a structured report, instead of having to integrate each RML processor separately.

Our contributions are: a report format for RML executions, an error taxonomy, RDF pointers for precise localisation, and a demo implementation showing a way to report the errors textually. We hope this proposal will spark a discussion in the RML community about improving the developer experience around RML mapping feedback.

## 2. The Challenge of Opaque Execution

Many RML processors are emerging, with processors specialised for different environments–from in-memory processing to streaming and distributed computing [3]. While this specialisation is necessary to address varying use-cases and thus (performance) requirements, it currently comes at the cost of interoperability due to different configuration and feedback systems. Ideally, an RML processor should function as a modular component in a pipeline: a black box that accepts (mostly) standardised inputs, including the special configuration and (mostly) standard RML mapping, and produces standardised outputs (including RDF datasets and feedback).However, while the mapping language uses a community-standard, the feedback output is not represented in a common representation. This lack of "standardisation" prevents processors from being interchangeable components.

We observed the following barriers:

- **Inconsistent reporting:** Processors report errors via channels–standard output or log files, or do not report anything and assume no RDF to be generated when errors are encountered.[3]
- **Silent fail:** A processor may produce an empty output without explicit errors. This non-trivial behaviour forces the user to guess whether the issue lies in the data, the mapping, or the RML processor configuration.
- **Contextual disconnect:** Errors are rarely contextualised within the original mapping document or data that is the cause of the error. But we can find a form of **limited contextualisation** in some processors (see Figure 1), which may include relevant mapping fragments, the specific data causing the issue, or suggestions for correction.
- **Terminology mismatch**: Errors are reported in processor-specific jargon instead of RML mapping semantics. Users to infer how the error message relates to the mapping may need knowledge of the processor's internals.

These barriers have two consequences for the RML ecosystem.

First, they create what we call an *Abstraction Gap*. RML is designed as a declarative language to abstract away the complexity of imperative data processing. However, when processors report errors using implementation-specific details (e.g., a Java's `NullPointerException` or Python's `index out of range`) instead of RML semantics, they exhibit a *Leaky Abstraction*. This forces the user to bridge the *Gulf of Evaluation* [4], the mental effort required to interpret the system's raw failure state in terms of their high-level mapping intent, rather than debugging the mapping. The user is forced to debug the processor's execution path, effectively breaking the declarative promise of RML and raising the barrier to entry for non-experts.

---

[3]Earlier versions of compliance coverage assumed that RDF would not be generated in the event of problems, leading to misleading representations of a processor's compliance when they relied on error codes.

```
11:50:05.836 [main] ERROR be.ugent.rml.cli.Main.run
    (442) - RER-scripts/resources/test-cases/rml
    -core/RMLTC0002e-JSON/student2.json
11:50:05.844 [main] ERROR be.ugent.rml.cli.Main.run
    (478) - RER-scripts/resources/test-cases/rml
    -core/RMLTC0002e-JSON/student2.json
java.nio.file.NoSuchFileException: RER-scripts/
    resources/test-cases/rml-core/RMLTC0002e-
    JSON/student2.json
    at java.base/sun.nio.fs.WindowsException.
        translateToIOException(
        WindowsException.java:87)
    ... 16 more
    at be.ugent.rml.cli.Main.main(Main.java:49)
java.nio.file.NoSuchFileException: RER-scripts/
    resources/test-cases/rml-core/RMLTC0002e-
    JSON/student2.json
```

```
Traceback (most recent call last):
  ... 8 more
  File "RER-scripts/logs-collector/.venv/Lib/site-
    packages/morph_kgc/mapping/mapping_parser.
    py", line 475, in _validate_termtypes
    raise ValueError(f'Found an invalid subject
        termtype. Found values {
        subject_termtypes}. '
                f'Subject maps must be {
                RML_IRI}, {
                RML_BLANK_NODE} or {
                RML_RDF_STAR_TRIPLE}.')
ValueError: Found an invalid subject termtype.
    Found values {'http://w3id.org/rml/Literal
    '}. Subject maps must be http://w3id.org/rml
    /IRI, http://w3id.org/rml/BlankNode or http
    ://w3id.org/rml/RDFstarTriple.
```

```
java.lang.IllegalArgumentException: Illegal
    character in path at index 4: juan daniel
    at java.base/java.net.URI.create(URI.java
        :932)
    ... 6 more
    at burp.Main.main(Main.java:42)
Caused by: java.net.URISyntaxException: Illegal
    character in path at index 4: juan daniel
    at java.base/java.net.URI$Parser.fail(URI.
        java:2995)
    ... 12 more
Illegal character in path at index 4: juan daniel
System exiting with code: 1
```

**(a)** RMLMapper (Java) on RMLTC0002e-JSON  **(b)** Morph-KGC (Python) on RMLTC0004b-JSON  **(c)** BURP (Java) on RMLTC0019b-JSON

**Figure 1:** User feedback output of implementation-leaking error reported by RMLMapper, Morph-KGC and BURP. RMLMapper (a) reports a file not found error, Morph-KGC (b) a term type mismatch (subject map as literal), and BURP (c) a URI syntax error (invalid character). We use ellipses (...) to abbreviate the stack traces, and paths are abbreviated with RER-scripts.

Second, the absence of machine-readable feedback hinders *tooling interoperability*. Without a shared error vocabulary, high-level tools (such as visual editors or CI/CD pipelines) cannot reliably parse or display feedback from different processors. This results in tightly coupled "tooling silos" where users are locked into a specific processor, not because of its performance characteristics, but because it is one that their debugging environment understands.

To resolve this fragmentation without sacrificing the performance benefits of specialised processors, we need a common representation for reporting execution outcomes. We argue that RML processors should produce not only the RDF data output but also a structured execution report.

## 3. Proposal: The RML Execution Report (RER)

To enable contextualised debugging and ensure uniform feedback across the RML ecosystem, we propose the RML Execution Report (RER), a formal, machine-readable vocabulary for describing the outcomes of an RML mapping execution. While the ultimate consumer of debugging information is the human developer, providing this data in a structured format is a prerequisite for building robust, interoperable tooling that relies on machine-readable output rather than fragile log parsing. Just as modern Integrated Development Environments (IDEs) rely on structured protocols like the Language Server Protocol (LSP) and the Debug Adapter Protocol (DAP) to provide rich editing and debugging experiences, RER empowers RML editors, dashboards, and CI/CD pipelines to visualise execution logs and metrics in the manner best suited to their users. While the relevance of such protocols have been demonstrated on the Semantic Web domain [5], standard protocols such as LSP or DAP are not fully suitable for declarative domain-specific languages (DSL) such as RML [6], necessitating a specialised vocabulary.

The RER vocabulary defines two primary components embedded within the report graph:

1. **Contextualised Structured Logs:** Log entries typed according to a taxonomy of errors and warnings linked, enriched with contextual metadata, for example, pointers to mapping definitions or incriminated data iterations.
2. **Execution Metrics:** Metadata regarding the run's success, completeness, and generation volume.

Whenever relevant, the errors are defined in terms of RML constructs (e.g., Logical Sources, Term Maps) rather than processor-specific internals, to help unify the error terminology and enable tools to traverse semantic links directly from the error instance back to the specific nodes in the mapping definition graph.

This structure representation will also support workflows, such as CI/CD pipelines that track regression or produce dashboards, thanks to metrics provided in the report.

The report is intended to be generated alongside the standard RDF output datasets. The RML Execution Report is itself expressed in RDF to guarantee semantic interoperability and align with the principles of RML, ensuring that errors carry precise, machine-interpretable definitions shared across all tools. Compliant processors may expose the report in a manner consistent with their architecture, whether materialised, exposed virtually, or emitted as a stream.

While the RER proposes the minimal *content* and *structure* of the feedback, the specifics of the human-readable rendering are left to individual processors or external tools. This separation of concerns allows each tool to specialise the presentation layer. For example, a CLI tool might render a text summary, while a GUI editor might highlight the error directly on the mapping definition.

We present a demo implementation based on BURP [7] later in this paper.

### 3.1. Conceptual Overview

### 3.1.1. Contextualized Structured Error Logs

The most important aspect of RER is the transition from unstructured text logs to semi-structured error data and information. Suppose an error occurs during the evaluation of a Reference Formulation. In that case, the report should capture not just the message but also the specific iteration and the mapping component involved. In RER, log entries are captured as a structured entity with a semantic *error type* from the RER taxonomy (see section 4), and the context is :

- **The Data:** The specific iteration or input value that caused the failure.
- **The Mapping:** The specific mapping statements responsible for the execution, when available.

And they can also include suggestions for remediation.

RER does not impose a strict ordering of logs, as distributed or parallel execution environments may make global ordering infeasible. Implementations may rely on timestamps rer:timestamp or an explicit ordering properties rer:order where applicable.

### 3.1.2. Execution Metrics and Metadata

To support pipeline observability, RER provides a set of suggested execution metrics. It allows RML processors to expose standard metrics while retaining the flexibility to define their own.

The report includes two status flags. First, rer:isSuccess acts as the semantic equivalent of the process exit code, but with inverted logic: a value of `true` corresponds to exit code 0, explicitly confirming that the execution logic completed without fatal errors. Second, rer:isComplete indicates whether the output dataset is guaranteed to be exhaustive. This distinction is particularly relevant for streaming processors, which may successfully process a given window of data (Success) but can only certify completeness once all input streams are closed and no data loss has occurred.

Beyond these flags, metrics can be included. Regarding the production volume of statements, RER defines the global rer:generatedStatements metric to report the total number of RDF statements.

For a more granular view, we suggest counting the *number of statements generated per RML triples map* (rer:generatedStatementsPerTriplesMap. This metric tracks the number of statements produced by a specific RML triples map. It should be noted that the sum of these granular counts is not necessarily equal to the global rer:generatedStatements. Downstream operations (such as deduplication of RDF statements or concatenation of RDF lists, as specified in RML-CC) may alter the total number of generated statements relative to the sum of intermediate yields. Furthermore, as different implementations may perform deduplication at different stages, this granular count may vary between RML processors for the same mapping and data.

Despite these limits, we believe that exposing these intermediate counters is valuable for:

1. **Silent Logic Bugs:** It facilitates the detection of "dead" or underperforming Triples Maps. If a processor reports success but a specific map generates zero or an unexpectedly low number of statements, it helps developers more quickly identify over-restrictive logical source filters or empty sources.

2. **Cardinality Verification:** When the user has specific expectations, this metric allows for a quick first validation of the dataset.
3. **Bottleneck Identification:** It helps identify "heavy" triples maps. While this metric does not directly measure execution time, identifying Triples Maps that generate disproportionately many statements provides a heuristic for targeting performance optimizations.

## 4. The RER Vocabulary and Taxonomy

The RER ontology defines a vocabulary for reporting RML mapping execution outcomes in a structured, machine-readable format. Developed to facilitate interoperability between RML processors and tooling, the ontology is available at https://w3id.org/dre/rer, with sources hosted at https://github.com/jduchateau/RER. The documentation is generated with Widoco [8]. The vocabulary uses OWL 2 to ensure consistency with the RML ontology, yet it relies on subclassing only and does not require reasoning capabilities.

The RER report is structured as an instance of type rer:RmlExecutionReport, containing the execution metrics defined previously subsubsection 3.1.2 and processor metadata such as name and version (rer:processorName and rer:processorVersion), as illustrated in Figure 3. The remainder of this section details the RER error taxonomy and the mechanism for precise error localisation using RDF pointers.

### 4.1. Error Taxonomy

To provide actionable feedback and facilitate automated analysis, we categorise errors into a hierarchy by problem domain. As it is impossible to anticipate solutions, the ontology is cause-oriented rather than solution-oriented, even though it tries to suggest both solutions and causes for a problem.

We suggest the following taxonomy for errors, based on the source of the problem (which part of the input):

- **Configuration Errors**: Errors related to the execution environment rather than the mapping logic or data content. These are external factors such as network timeouts, file permission issues, authentication failures, or insufficient memory
- **Mapping Errors**: The artefact is ill-formed (e.g., syntax errors, unsupported functions). These include syntax errors, invalid references, violations of the RML specification (e.g., missing required properties), and logical issues such as circular joins. They could mostly be checked by a static analysis of the mapping graph, without data.
- **Execution Errors**: The mapping is valid, but the runtime environment or data caused a failure (e.g., network timeout, unreachable source). Subtypes of execution errors include:
  - **Data Errors**: The data does not conform to the expectations set by the mapping (e.g., type conversion failures).

These top-level error domains are not disjoint. For example, in RML-FNML, a not-found function may be typed as a mapping error when it was specified as a constant or as an execution error when defined with a template. Such an error can also be a configuration error if the function is not available by default to the processor, but could have been provided by the mapping author.[4] In these cases, use the rer:UnsupportedFunction with what fits best among the domains.

We also included classes that represent errors that may pertain to multiple domains or do not clearly fit in any such domain. These "artificial" classes, such as rer:ReferenceFormulationError or rer:DatatypeExpectationError, are thus used to group specific errors that are related to facilitate tools in generalising their handling.

---

[4]MorphKGC, for example, allows one to dynamically include custom functions with decorators.

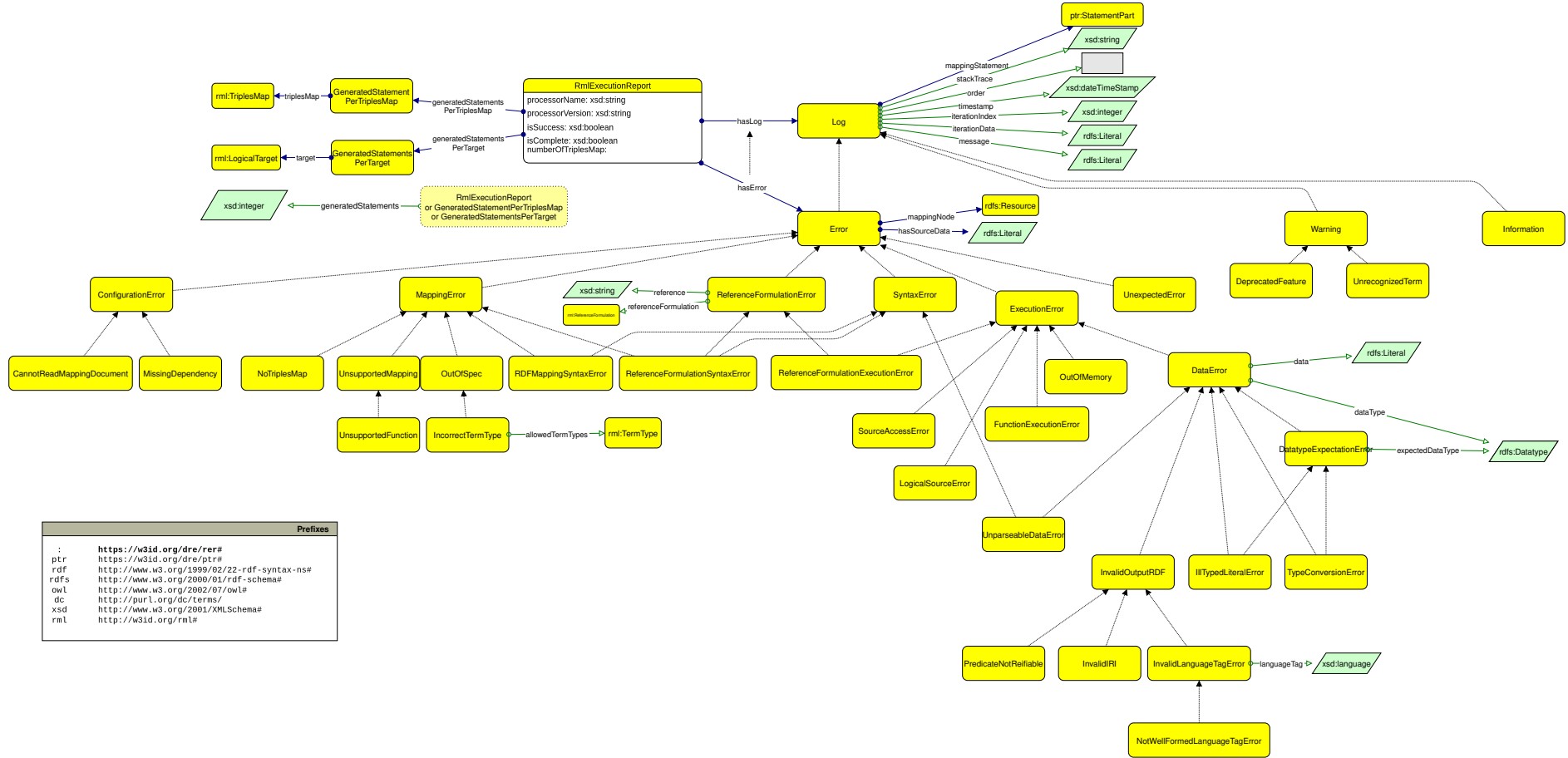

**Figure 2:** RER ontology and error taxonomy overview. Using a mix of UML and Graffoo visual languages. Created using draw.io and a custom plugin to import the ontology (accessible at https://github.com/jduchateau/graffoo-import-drawio-plugin).

## 4.2. Precise Localisation with RDF Pointers

In the context of debugging an RML mapping, some errors need a higher level of granularity than pointing to a named node. We introduce the concept of a *statement part* to address this. A pointer can identify the Subject, Predicate, or Object of a generated triple. Furthermore, if the error occurs within an RDF Literal (e.g., a datatype conversion error, an invalid language tag, or a reference formulation syntax error), the pointer can target specific components of that literal, or a range of characters in the literal's value. These RDF Pointers are in their own namespace https://w3id.org/dre/ptr and enable pointing to a specific part of the RDF graph.

One common use case is to give a RML mapping file manually authored in a textual format (e.g., Turtle or YARRRML[5]) to a RML processor. To support that use case effectively, we must be able to "translate" the RDF Pointer to its location in the file. Such translation may lead to multiple physical locations within the file. Our implementation, described in the next section, supports this for Turtle files.

```
:report a rer:RmlExecutionReport
        rer:isSucess false;
        rer:isCompleted false;
        rer:generatedStatements 0;
        rer:generatedStatementsPerTarget [
                rer:target rml:Default;
                rer:generatedStatements 0;
        ];
        rer:generatedStatementPerTriplesMap [
                rer:triplesMap <PersonTM>
                rer:generatedStatements 0
        ];
        rer:generatedStatementPerTriplesMap [
                rer:triplesMap <MovieTM>
                rer:generatedStatements 0
        ];
        rer:hasErrors [
                rer:timestamp "2025-11-26+13:20:00"^^xsd:dateTime
                a rer:ReferenceFormulationExecutionError
                rer:message "Execution Error in JSONPath `$.film.title.value`."
                rer:reference "$.film.title.value"
                rer:referenceFormulation rml:JSONPath
                rer:iteration "{film:{title:'A film title'}}"
                rer:stackTrace "[[omitted a java stack trace]]"
                rer:mappingStatement [
                        rer:statement <<:a0 rml:reference "$.film.title.value`.">>;
                        rer:part rer:object
                ];
        ]
.
```

**Figure 3:** Example of a structured RER Error (Turtle) with RDF Pointers. The error is an execution error in a JSONPath reference formulation, due to the data not having a `value` property.

# 5. Implementation and Validation

To validate the feasibility of the RML Execution Report, with the development of a Proof of Concept (PoC) implementation. We chose to fork the BURP [7] RML processor because it is an implementation with no optimizations and, as such, its source code reflects the concepts of RML. The goal of this PoC was not to produce a production-ready processor, but to demonstrate that rich, structured error reporting is possible. The source code is available at https://github.com/jduchateau/BURP-Errors or on https://doi.org/TODO_FOR_CAMERA_READY.

---

[5]YARRRML Specs: https://rml.io/yarrrml/spec/

## 5.1. Implementation of the RER-enabled RML Processor

The primary challenge in implementing RER is maintaining the link between the runtime execution and the static mapping definition. Standard RML processors typically parse the mapping into internal data structures and discard the original location within the RML mapping graph. To address this, we modified the BURP parser to retain provenance information. Specifically, we extended the internal model classes to include the origin information of the mapping definitions. To link back to the original turtle file, we used a custom RDF parser that exposes token positions in the RDF graph.

During the execution phase, we instrumented the pipeline to:

1. **Accumulate Statistics:** We verify the completion status and count generated statements per logical target on the fly.
2. **Catch and Contextualise Exceptions:** Instead of allowing exceptions to propagate up to a generic handler, we intercept them at the level of the `TriplesMap` or `TermMap` execution. This allows us to wrap the raw exception (e.g. a parsing error or network timeout) into a structured RER error instance, enriched with the retained mapping metadata.

## 5.2. Human-Readable Output Generation

While the internal instrumentation described above ensures that errors are captured with high fidelity in the RER graph, this RDF data is not immediately useful to a user of RML processors debugging in a terminal. Users of RML processors often interact directly with them via a command line interface (CLI). In this context, presenting the RER graph directly imposes a high cognitive load: users would be forced to mentally parse the RDF structure to identify simple errors. Therefore, converting this structured data into a human-readable textual rendering is critical for usability.

As discussed in the proposal section, the RER specification deliberately separates the error *data* (the RDF graph) from its *presentation*. Our implementation demonstrates this principle by including a specific rendering module for the CLI. This module acts as a view layer: it consumes the generated RER graph and adapts it to the textual metaphor of the command line. By leveraging the RDF Pointers contained within the report, the module reconstructs the context of the error, highlighting the precise snippet of the mapping definition responsible for the failure.

The following listing listing 2 shows an example of textual output generated by our PoC. It is an instance where we try to align the rendering of RER graph feedback with the textual *metaphor* and the user's *mental model* of the mapping document by projecting back the errors onto the source code.

**Listing 1:** Mapping of RMLTC0023a-JSON

```
1  @prefix foaf: <http://xmlns.com/foaf/0.1/> .
2  @prefix rml: <http://w3id.org/rml/> .
3
4  <http://example.com/base/TriplesMap1> a rml:TriplesMap;
5    rml:logicalSource [ a rml:LogicalSource;
6        rml:referenceFormulation rml:JSONPath;
7        rml:iterator "$.students[*]";
8        rml:source [ a rml:RelativePathSource;
9            rml:root rml:MappingDirectory;
10           rml:path "student.json"
11         ]
12     ];
13   rml:subjectMap [
14       rml:template "http://example.com/{{Name}}";
15       rml:class foaf:Person;
16     ] .
```

**Listing 2:** Error output of our implmentation for RMLTC0023a-JSON, notice the the higlight inside the template literal

```
Errors:
  Syntax error in JSONPath `{Name` at 1:0: token recognition error at: '{'
    In mapping
      resources/rml-core/RMLTC0023a-JSON/mapping.ttl:14:41-14:45
```

```
 13 |    rml:subjectMap [
 14 |        rml:template "http://example.com/{{Name}}";
 15 |        rml:class foaf:Person;
   is a Reference Formulation Syntax Error (http://w3id.org/rml/report#ReferenceFormulationSyntaxError)
      | An error indicating that the reference formulation expression in the mapping has invalid syntax.
      | Try to: Correct the reference formulation syntax according to the rules of the chosen reference formulation.

Statistics:
  - Number of triples maps: 1
  - Generated statements: 0
  - Generated statements per triples map:
      * http://example.com/base/TriplesMap1: 0
```

This PoC implementation illustrates the feasibility of moving beyond the opacity of exit code 1 to actionable and in context feedback. By projecting errors back onto the mapping definitions, it aims to provide the semantic context necessary to help users bridge the Gulf of Evaluation between the runtime failure and their original intent.

# 6. Related Work

The challenge of debugging declarative data transformations is not unique to RML. To improve on the lack of standardised feedback in RML, we draw inspiration from established error reporting mechanisms in related domains: database query systems, ETL pipelines, and general-purpose programming languages. Each of these fields faces the challenge of debugging data transformations and has developed mature patterns for observability that RML currently lacks.

In relational databases, the SQL standard [9, sec. 24] defines status codes accessible with the SQLSTATE command. These codes are classified in categories (Success, Warning, No-Data, and Exception), Class, and Subclasses. Moreover, to get more context in addition to the error kind, the norm defines the GET DIAGNOSTICS command [9, sec. 23] that provides structured details such as error messages or row counts of the last SQL statement. Beyond these standardised reports, RDBMS implementations often provide diagnostics like EXPLAIN and ANALYZE to expose the query execution plan and performance metrics. However, these commands are not part of the standard because they reveal internal optimisation strategies specific to the RDBMS's implementation, analogous to how an RML execution plan would depend heavily on the specific architecture of the RML processor.

In general-purpose languages, error handling is a core part of language design. Languages like Java use a Throwable class hierarchy, distinguishing between recoverable Exceptions and fatal Errors. Unlike SQL states, these are rich objects carrying metadata, execution stack traces, and nested causes (chained exceptions).

To address the interoperability of static error reporting across tools, the *Static Analysis Results Interchange Format* (SARIF) [10] provides a standard JSON schema for static analysis. SARIF enables disparate tools to report results to common viewers (like IDEs) by standardising the representation of file locations and code snippets, with physical and logical locations as well as regions for text, binary. RER adapts this philosophy of interoperable, specifically for RML, with location indication specifically adapted to the RDF model instead of JSON. We were made aware of SARIF lately in the process, and maybe it could be interesting to extend it or provide a mapping with it, could provide larger interoperability, at the expense of semantic interoperability.

In the programming domain, *algorithmic debugging* (or declarative debugging) has been explored. Examples include [11] for lazy functional languages and [12] for C, these approaches rely on an oracle (the user) to validate execution steps based on externally visible symptoms to isolate the fault. Current RML tooling, however, lacks the feedback mechanisms required to support such interactive or post-mortem analysis.

In the declarative domain, XSLT [13, sec. 2.14] explicitly distinguishes between *static errors* (detected during stylesheet analysis) and *dynamic errors* (runtime failures). This distinction is similarly reflected in RER, which separates mapping errors from execution errors.

Furthermore, XML pipeline languages like XProc defined error handling, allowing users to define

`try/catch` steps within the pipeline itself and error vocabulary (sec. 4.6.1) to specify a format for errors (including an error code and the step causing the issue and a physisical location, ). While RER focuses on *reporting* rather than control flow, it is another domain where RML could draw inspiration for future extensions.

In the RML ecosystem, the RML core specifications only specify data error like R2RML before it. RML-SHACL allows for the validation of the mapping document structure itself, ensuring syntactical correctness but not catching runtime data contradictions. RML Test Cases provide a compliance suite but rely on binary success/failure status codes, which define broad conformance but offer little aid in diagnosing specific failures. Regarding provenance, RMLMapper [14] supports the generation of PROV-O traces. However, RMLMapper does not expose mapping feedback in an interoperable way: it can emit user-facing messages, but these are not presented back to the user with line/column highlights inside the RML mapping document as our PoC does for Turtle.

RER aims to bring structured, contextualised error reporting to the RML ecosystem. Inspired by the hierarchical classification of SQL and the interoperability of static analysis tools, RER adapts these concepts to the RDF nature of RML and takes advantage of semantic technologies to define an error taxonomy.

## 7. Discussion and Next Steps

Improving the Developer Experience (DX) of Knowledge Graph Construction requires more than just faster processors and a more expressive RML language; it demands a robust ecosystem of intelligent tooling (from visual editors to automated debuggers). However, the development of such an ecosystem is currently stalled by the lack of interoperability between the underlying RML processors and the upper-layer tools. RER addresses this bottleneck by establishing a shared contract for status and error reporting. In this section, we discuss how this contract enables the shift from simple signalling to deep diagnosis, the architectural opportunities it reveals.

**Beyond Binary Exit Codes** The RML community and its challenges currently rely on binary exit codes. While a binary exit code provides a necessary signal to show that a failure occurred (in CI/CD pipelines or RML tests themselves), it is insufficient for diagnosis or debugging. Recent community discussions[6] have explored expanding this system into a standard list of status codes. However, we argue that exit codes constitute an insufficient information signal; they indicate that a problem, or more, of a given class occurred, but fail to convey *where* or *why* it happened, nor do they provide *actionable feedback* for remediation.

By using RML-specific semantics instead of implementation-specific error definitions, RER shifts the burden of interpretation from the user to the processor. Instead of forcing the user to reverse-engineer the processor's internal state, the processor explicitly attributes the failure to the RML mapping. In terms of feedback understandability, this bridges the *Gulf of Evaluation.*

By adopting RER as a shared contract, tools need only implement a single RER consumer to support any compliant processor. This decouples error generation from error presentation. Downstream agents can adapt the machine-readable report to the specific metaphor of the tool (e.g. CLI or GUI), aligning the feedback as closely as possible with the user's mental model.

**Error Recovery and Handling** Going further, Nielsen's heuristics [15] state that error messages should not only report the problem but also offer a recovery path. While the current RER proposal includes static suggestions for remediation, future work could investigate context-specific remediation to help resolve failures. Beyond reporting, there is also an opportunity to integrate error handling directly into the mapping lifecycle. Future RML modules could support control flow constructs (e.g., `try/catch` blocks) that define fallback logic for specific RER error classes. This would allow users to

---

[6]including https://github.com/kg-construct/rml-core/issues/32

proactively handle expected data inconsistencies (e.g., `Catch rer:DataError Use Default`) within the mapping itself.

**Implementation Insights and Limitations**   Implementing RER highlights specific engineering challenges regarding provenance. As observed in our PoC (BURP-Error), a primary hurdle is that many processors parse the RDF mapping graph into native data structures in a *lossy* manner, discarding the link to the original graph nodes during transformation. To support RER's RDF Pointers, processors must maintain a link/lineage between the execution structures and the original mapping definition. Our partial review (see section A) suggests that RML processors rely on imperative parsing or monolithic SPARQL queries, lacking a modular, declarative approach to parsing RDF graphs into objects while preserving their origin.

Furthermore, this increased observability may introduce overhead; however, measuring it is left for future work. Tracking token positions and maintaining links to the mapping graph adds memory and processing overhead. This creates a tension between the goals of high throughput (production) and deep observability (debugging). Balancing these conflicting requirements remains an open architectural challenge; future processors may need to explore configurable observability levels. However, this performance awareness also opens new debugging possibilities: extending RER to include processor-specific *Execution Plans* would allow users to understand performance bottlenecks alongside logic errors.

Additionally, our experience highlighted that monolithic architectures increase the integration effort for such reporting mechanisms. A more modular design would facilitate the injection of error reporting hooks and future RML experimentation.

Finally, while the raw RDF report is ideal for machine exchange, it imposes a high cognitive load on humans. Thus, a dedicated presentation layer (such as the textual renderer in our PoC) remains essential for CLI users. While RER requires development effort, the value lies in enabling a common interface for all upper-layer tools.

**Future Directions**   The next step is the validation of RER through broader adoption and collaboration with other engine maintainers. We also suggest modularising the errors vocabulary and aligning those modules with the RML modules. Each RML module would contain its own error module, e.g., 'RML-IO-Error', with the 'RML-Core-Error' module containing the common errors and the report structure. Compliant RML processors with 'RML-Errors' would then just have to comply with the associated error module of the RML modules they support. However, looking forward, we believe the most transformative step will be moving from post-mortem analysis to *interactive debugging*. We envision a mapping tool that would include debugging capabilities. This would evolve the RML ecosystem to an interactive development environment, allowing users to pause, inspect, and step through executions as they happen.

## Acknowledgments

This work was supported by the Fonds de la Recherche Scientifique – FNRS under Grant n° MIS F.4016.24.

## Declaration on Generative AI

During the preparation of this work, the author(s) used Gemini 3 in order to: Grammar and spelling check, Paraphrase, and reword. After using these tool(s)/service(s), the author(s) reviewed and edited the content as needed and take(s) full responsibility for the publication's content.

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

# A. RER Taxonomy and Report, Design methods

To ensure the RER taxonomy is sound at the RML abstraction level and covers practical realities, we derived the taxonomy of errors from multiple experiments, not necessarily in order.

**Mental Model Construction**   We analyse errors in RDF triples generation based on RML-Core Specs sections 12.1 and 12.2 [2] (Generated RDF Term of a Term Map). Through a thought exercise, we identified potential failure points in RML processors and categorised errors based on the type of solution required. This solution-oriented categorisation, while similar to a cause-based approach, focuses on actionable resolutions.

**RML processor Code Analysis**   Theoretical models may miss the nuances of an actual implementation. Therefore, we performed a manual analysis of the BURP codebase [7]. With a static analysis of the code, we extracted all thrown exceptions and error handling blocks. This revealed low-level errors and errors from other RML modules implemented by BURP. We also observed that the categorisation is not purely exclusive, a mapping error can sometimes also be a configuration error, or execution errors can be mapping errors when constant.

**Test Case Validation**   We analysed the 56 expected failing test cases from the RML test suites (IOREG, TC, STC, LVTC, FNMLTC) with the following RML processors: BURP[7], RMLMapper[14], Morph-KGC,[16] and Flex RML[17]. A short example is presented in the article at Figure 1.

**Specification Annotation**   We also tagged the RML-Core specification to identify implicit error states defined by the standard. This ensured that our taxonomy remained aligned with the specification's intent rather than overfitting to a specific implementation.