# OpenReview forum: "Beyond Exit Code 1: Execution Report of RML Processors"
_eswc-conferences.org/ESWC/2026/Workshop/KGCW — KGCW 2026_

### Official Review · ~Michael_Freund1 · 2026-04-02
**Strong and useful work, but the evaluation could be stronger**

**Rating:** 7
**Confidence:** 4

**Review:**

The paper introduces the RML Execution Report (RER), an RDF-based reporting format for RML processor execution. While structured error-reporting standards exist in other domains, applying a machine-readable reporting model to the RML ecosystem is a novel and relevant contribution. The topic fits the workshop well, particularly given its focus on improving tooling and developer support. The paper is well structured and clearly motivates both the problem and the way RER addresses it.

## Strengths
S1. Expressing RER itself in RDF makes the reports interoperable and grounded in a shared ontology. The proposed taxonomy and vocabularies also appear extensible, which should support future evolution.

S2. The approach decouples error generation in an RML processor from error presentation in downstream applications such as IDEs or CLIs. This design mirrors successful patterns from other ecosystems such as LSPs.

S3. The CLI rendering example demonstrates that RER can be transformed into actionable feedback for users without exposing them directly to raw RDF output.

## Weaknesses
W1. The evaluation is limited. The paper does not provide a quantitative assessment of taxonomy coverage or user-facing benefits. It also remains unclear whether the approach primarily benefits non-expert users, expert users, or both.

W2. The taxonomy is derived from manual analysis, but the paper does not show a systematic validation process, for example against a corpus of special error test cases. As a result, possible gaps, overlaps, or ambiguities in the hierarchy may remain unnoticed.

## Overall Assessment
The paper makes an original contribution to the RML community. Its central idea is practical and likely to improve the usability of RML tooling. The proposal is valuable because it treats reporting as a reusable interface rather than as a processor-specific implementation detail. The contribution has the potential for broad impact across both research prototypes and production-oriented KG pipelines.
In summary, the paper presents a original contribution with clear relevance and practical value.

---

### Official Review · ~Markus_Schröder1 · 2026-04-03
**Very relevant topic and a first well-written proposal**

**Rating:** 8
**Confidence:** 4

**Review:**

The paper introduces well the topic of a missing reporting format for RML processors. Section 2 explains in detail the typical challenges and shows in Figure 1 examples of real feedback output. The text gives a good motivation towards the proposed RML Execution Report. The vocabulary and taxonomy is well documented with a website, diagram, descriptions and an example RDF output. I really like the interactive debugging outlook at the end of the paper.
All in all, I think the topic is very relevant and the paper proposes a first good solution that should be adapted by RML processors in the future.

Some comments:

In Section 3 when "Structured Logs: Log entries" were mentioned, it was not clear to me if the RML Execution Report (RER) is intended to log the whole execution or just when an error occurs. This could be made more clear. Why is it not called RML Error Report? I guess because on a successful run, there is still not an empty RER. Looking at the ontology there is also an Information Log class.

Suggestion for 3.1.2. Execution Metrics and Metadata: "rer:generatedStatements" it might be possible to reuse the https://www.w3.org/TR/void/ vocabulary which has void:triples, void:entities. etc.

Suggestion for 4.2.: By refering to a statement with RDF-Star [cite], "... an [additional] pointer can identify the Subject, Predicate, or Object of a generated triple".

Minor:
* Page 2: "feedback).However," missing space.
* Figure 3's caption could mention that due to limited space stack trace is omitted and use "(...)" in ttl code.
* Language Server Protocol (LSP) and Debug Adapter Protocol (DAP) could have a footnote.

---

### Official Review · ~Pieter_Colpaert1 · 2026-04-04
**Interesting idea worth discussing**

**Rating:** 8
**Confidence:** 5

**Review:**

RML is actively being standardized at the W3C. This paper rightfully notes a gap between standardizing the mapping language, and the execution reports that has not received any attention so far. The work could be (but isn’t in the paper - which is fine) compared to SHACL validation reports (https://www.w3.org/TR/shacl/#validation-report), that is a first-class citizen of the SHACL spec, or to FORCE, the framework for evaluating ODRL policies (https://w3id.org/force), that also has been proposed in hindsight.

The paper is very well written and easy to follow. The resource the paper introduces is openly available (http://w3id.org/dre/rer) and follows state of the art best practices. I genuinly hope this idea gets larger support in the W3C KGC CG.

Minor:
 * Abstract: not sure what “with test batteries” means in this context.
 * I found the related work section far from the current paper’s topic.

---

### Decision · Program_Chairs · 2026-04-09

**Decision:**

Accept

**Comment:**

This paper has been selected for presentation at the KGC workshop. We strongly encourage the authors to consider the reviews whilst revising the paper. Camera-ready instructions will soon follow.